# *Galleria mellonella* as a Good Model to Study *Acinetobacter baumannii* Pathogenesis

**DOI:** 10.3390/pathogens10111483

**Published:** 2021-11-14

**Authors:** Ye Tao, Luminita Duma, Yannick Rossez

**Affiliations:** 1Université de Technologie de Compiègne, UPJV, UMR CNRS 7025, Enzyme and Cell Engineering, Centre de Recherche Royallieu–CS 60 319 , 60203 Compiègne, France; ye.tao@utc.fr (Y.T.); luminita.duma-catoire@cnrs.fr (L.D.); 2Université de Reims Champagne-Ardenne, CNRS, ICMR UMR 7312, 51097 Reims, France; 3Université Lille, CNRS, UMR 8576-UGSF-Unité de Glycobiologie Structurale et Fonctionnelle, F-59000 Lille, France

**Keywords:** *Acinetobacter baumannii*, *Galleria mellonella*, host–pathogen interactions, virulence factors, therapy strategies

## Abstract

The invertebrate model, *Galleria mellonella*, has been widely used to study host–pathogen interactions due to its cheapness, ease of handling, and similar mammalian innate immune system. *G. mellonella* larvae have been proven to be useful and a reliable model for analyzing pathogenesis mechanisms of multidrug resistant *Acinetobacter baumannii,* an opportunistic pathogen difficult to kill. This review describes the detailed experimental design of *G. mellonella*/*A. baumannii* models, and provides a comprehensive comparison of various virulence factors and therapy strategies using the *G. mellonella* host. These investigations highlight the importance of this host–pathogen model for in vivo pathogen virulence studies. On the long term, further development of the *G. mellonella*/*A. baumannii* model will offer promising insights for clinical treatments of *A. baumannii* infection.

## 1. Introduction

Over the past decades, *Acinetobacter baumannii* has widely emerged as one of the major causes of highly invasive nosocomial pathogen infections in the health system [1]. Infections by this microorganism are responsible for increased morbidity and mortality, and make a huge burden to patients and hospitals [2]. As the top concerning microorganism on the global priority list ranked by the World Health Organization (WHO) [3], *A. baumannii* is a multi-drug resistant (MDR) bacterium which needs new drug development [4]. Therefore, the screening of the most adapted animal models for studying pathogenic mechanisms and therapeutic strategies before clinical therapies is particularly critical.

A series of animal models have been examined and established for *A. baumannii* studies, including mammalian and non-mammalian models. Murine models [5] are still the predominant mammalian models in *A. baumannii* researches, though some other mammalian models have also been tested, such as rabbits [6], guinea pigs [7], and porcine models [8] (Figure 1a). *A. baumannii* is frequently associated with pneumonia, making small rodent lung infection models well suited for these bacteria [9]. However, increasing costs and growing ethical concerns made the use of rodents more difficult [10]. Non-mammalian models, such as *Galleria mellonella* (greater wax moth) [11], *Caenorhabditis elegans* (roundworm) [12], *Dictyostelium discoideum* (slime mold) [13], *Danio rerio* (zebrafish) [14] and *Drosophila melanogaster* (common fruit fly) [9], are also informative to decipher virulence factors needed during host–pathogen interactions of *A. baumannii*. Among them, *G. mellonella* caterpillars have attracted more and more attention in the last ten years (Figure 1a). The keywords for each node distributed in time-zone visualization (Figure 1b) indicate an increased interest towards the *G. mellonella* model system. The research involving *G. mellonella* model mainly focused on *A. baumannii* pathogenicity factors (such as surface antigen proteins and efflux pump) and drug therapies.

*G. mellonella*, also known as a wax moth, belongs to Lepidoptera order from the Pyralidae family. This moth is distributed worldwide, and is commercially available for fishing or to feed reptiles and birds, making them readily accessible [15]. The last larval stage of this insect has been utilized as a host model to extensively study in fungi pathogenesis, including *Conidiobolus coronatus* [16,17,18,19], *Beauveria bassiana* [20,21], *Metarhizium* species [21,22,23], and so on. Furthermore, bacteria, including gram-positive [24,25] and gram-negative [26,27], have also been utilized in studying with *G. mellonella* models. The benefits of using *G. mellonella* models are numerous. *G. mellonella* produce a huge progeny quantity with a short life cycle, and are inexpensive because they are easy to rear without special laboratory infrastructure. The possibility of using many animals per experiment makes them eligible for high-throughput studies. The relatively large size of the larvae (12–20 mm) allows precise quantification of the inoculation, and facilitates handling for tissue extraction and histological analysis [28,29]. Importantly, there is no ethical approval requirement for research on *G. mellonella* [30].

Despite a large number of articles describing the feasibility and safety of *G. mellonella* for microbial studies [31], its value for drug-resistant microorganisms remains to be explored. In this review, we highlight why *G. mellonella* can be used as a model for MDR *A. baumannii* infection, the contributions of this model to study *A. baumannii* pathogenicity, and to target the most effective and prospective therapy strategies to fight *A. baumannii* infection.

## 2. *G. mellonella*-Based Model

*G. mellonella* has a rapid life cycle with four developmental stages: egg; larvae; pupa; and adult moth [32] (Figure 2a). Differences in temperature and humidity affect the developmental speed, with a full life cycle under favorable conditions being only 8–12 weeks [33]. The white dome-shaped eggs hatch to larvae in about 1–2 weeks at 28–34 °C [33]. The creamy-colored larvae pass through 8–10 molting stages in 5–6 weeks until cocoon development [33]. After 2–3 weeks of incubation, the reddish-brown pupa evolves into a pale cream moth [33].

### In Vivo Model

Insects’ innate immune system has been well documented to protect them against infection from a broad spectrum of pathogens [34]. Genome research has shown that larvae have many homologous genes to humans, who participate in pathogen recognition and signal transduction [35]. In *G. mellonella*, the innate immune system is constituted by cuticle, cellular, and humoral immune defense [36].

The cuticle represents the first line of protection, and is mainly composed of chitin, lipids, and protein matrices. All of these molecules act as a physical barrier to prevent invasion by pathogens [36]. The cuticle is organized in three outer layers, including the epicuticle, procuticle, and epidermis [31] (Figure 2b). The intact epidermis prevents pathogen entry, but once it breaks down due to damage or degradation, cellular and humoral immunity take over the defense.

The cellular immune system is mediated by phagocytic cells, called hemocytes, which are mainly responsible of encapsulation, nodulation, and phagocytosis [30,37] (Figure 2b). To date, six out of the eight types of hemocytes found in insects have been identified to be responsible of these functions in *G. mellonella* (plasmatocytes, granulocytes, prohemocytes, spherulocytes, coagulocytes, and oenocytoids) [28,38]. Firstly, granular cells attack the penetrated microorganisms, then, the process promotes the attachment of plasmatocytes to form a layer of cells, resulting in encapsulation and nodulation. Phagocytosis is similar to human cellular defense reactions with the participation of hemocytes [31]. The humoral immune response is highly regulated by soluble effectors, such as complement-like proteins (opsonins), melanin, and antimicrobial peptides (AMPs), which play a role in melanization, hemolymph clotting, and primary immunization [39] (Figure 2c).

In the early stage of *A. baumannii* invasion, the larval immune response is activated, and struggles against *A. baumannii* virulence factors. If the infection is controlled by the immune system, the larvae will survive—alternatively, the larvae will continue melanization and finally die. The two different responses are dependent of the phagocytosis by hemocytes, or the melanization caused by the deposition of melanin around microorganisms [40].

## 3. Experimental Design Suitable for *G. mellonella*/*A. baumannii* Interaction

Generally, the larvae are employed at the 5th to 6th instar, at about 2–3 cm length and a weight of around 250–350 mg. The spontaneous mobility of larvae is a good indication of their viability [33,41]. For one experiment, the larvae are conventionally divided into three groups of about 10 to 20 individuals, one group inoculated with PBS, one group with bacteria sub-divided by the different conditions/strains needed, and one group without injection. In Table 1 and Table 2, the inoculation methods, culture conditions, and larval detection indicators are listed. These different studies have described virulence factors of *A. baumannii* (Table 1) and antimicrobial agents tested against *A. baumannii* (Table 2) in *G. mellonella*.

After 24 h of starvation at room temperature, three inoculation methods have been described to work with *G. mellonella*: topical application [97]; force-feeding [98]; and injection [11]. For *A. baumannii* infection, only the injection method into the hemocoel of the larval cuticle of the last left proleg [40] has been used (Figure 3a). For drug treatment, the correct timing of drug administration is also important, commonly within 3 h after *A. baumannii* injection. In some studies, drug application before or simultaneously with *A. baumannii* infection has been reported, but such cases are rare [89,91,94]. Compared to the two other methods, the injection has the advantage to accurately deliver the inoculum, and is therefore more reproducible [40]. However, the control group, injected only with buffer or medium, is crucial to ensure that the death of larvae is not caused by trauma or solvents.

*G. mellonella* larvae can be maintained at different temperatures after injection, between 15 °C to over 37 °C [99]. In order to better understand the interaction between the host and the pathogen in an environment closer to the mammalian organism, 37 °C is the most employed temperature for *A. baumannii* infection [29]. The viability, motility, and virulence of *A. baumannii* at 28 °C [65] and 30 °C [40] were also studied in order to assess the adaptability of the different clinical strains’ response to environmental changes. The incubation duration inside the larvae usually varies from few hours to few days. Experiments suggest that too short periods (<4 h) are not conducive to an accurate evaluation of *A. baumannii* virulence or drug efficacy. Conversely, after too long (>8 days) time periods, the larvae metamorphose into moths.

The *G. mellonella* larvae assessments could be larval mobility [90], mortality/survival rate [72], histological analysis [11], and bacterial numbers recovered after incubation [64]. Table 3 introduces the health index scoring system to evaluate the larval health status, including larvae mobility, cocoon formation, melanization, and survival [99]. The movement, observed by touching and the melanization, visible by naked eyes, are keys to distinguish the larval morbidity after *A. baumannii* infection (Figure 3b) [90]. Though the *A. baumannii* virulence overcomes the larval immune system over time, the larval movement gradually decreases, and the melanization progresses gradually. Complete melanization indicates death. Mortality/survival rate is the most monitored indicator, which directly reflects *A. baumannii* virulence. The survival percentage, usually characterized by the Kaplan–Meier curve, is investigated every 24 h [100]. Histological analyses are essential for studying host–defense mechanisms and pathogen infection pathways. A rare study associated with tissue damage, fat body, and muscle layer melanization has been reported for *A. baumannii* infected larvae [11].

## 4. *A. baumannii* Pathogenicity in *G. mellonella*

The study of *A. baumannii* pathogenesis is critical to provide a theoretical basis for the development of new therapeutic modalities and drugs. Many studies have documented the pathogenic mechanisms of *A. baumannii* infection by using *G. mellonella* models, and here, we will focus on the virulence factor studies, the antibiotics resistance mechanisms, and finally, we will discuss the *A. baumannii*’s persistence in a broad range of environments/hosts.

*A. baumannii* pathogenicity has been decoded in part with the help of *G. mellonella*. For example, the impact of the phase variation on *A. baumannii*‘s virulence was performed on this model. This variation corresponds to the transition between opaque and translucent colonies [101]. For *A. baumannii*, opaque variants are more virulent in larvae models, whereas translucent variants have the ability to form more biofilm [67].

### 4.1. A. baumannii Virulence Factors and G. mellonella

The ability of *A. baumannii* to persist in many circumstances, and to be life-threatening, is partly due to its virulence factors. The recognized *A. baumannii* virulence factors studied with *G. mellonella* are relatively scarce.

Phospholipases (PLs) can lyse the host-cell membrane by catalyzing the hydrolysis of phospholipids to facilitate bacterial invasion [5]. So far, two PLs have been identified in *A. baumannii*: the phospholipase C (PLC) and the phospholipase D (PLD) [102]. PLC cleaves and releases the phosphorylated head group from phospholipids, whereas PLD cleaves off only the head group [103]. Both cut between the phosphorylated and the polar head groups. In addition, PLs can disturb the host–immune response by generating second messengers such as phosphatidic acid, which can also promote the pathogenesis [104]. Kareem et al. have tested 30 *A. baumannii* strains collected from hospitalized patients with *G. mellonella* killing assays [42]. The results clearly showed a higher larvae mortality when PLC is combined with elastase (lasB), a virulence factor that has the ability to degrade host tissue. Fiester et al. have shown the existence of two PLC (PLC1 and PLC2), but only PLC1 appears to play a critical role during *G. mellonella* infection [43]. In the case of PLD, Stahl et al. have found three different PLD needed in a concerted manner to successfully infect *G. mellonella* [44].

Only a few membrane proteins have been identified as virulence factors in *A. baumannii*: the outer membrane protein A (OmpA) [105] and 33 (Omp33) [106] are the best characterized, but have not been investigated with *G. mellonella*. They are able to adhere to the epithelial cells of the host, leading to biofilm formations that contribute to the invasion of *A. baumannii,* and thus, to the apoptosis induction of the host cells. However, surface antigen protein 1 (SurA1) was examined in larvae models to evaluate the virulence of *A. baumannii* strains [45]. It was found that the SurA1 knock-out mutant displayed a lower fatality rate in larvae infection assays, suggesting the importance of SurA1 in *A. baumannii* virulence [45].

The capsular polysaccharides (CPS) of *A. baumannii* are made up of oligosaccharides (K units) with various carbohydrate types, varying in numbers and type of linkage, and with acetyl, pyruvyl groups, or other modifications [107]. The CPS genes are located in the K locus, and are positively correlated with the virulence of *A. baumannii* [108]. Over 100 unique capsule loci have been identified in *A. baumannii* to date [109]. Xu et al. have interrupted *gnaA* (a gene found in the K locus), and tested the pathogenicity of the resulting strain through a larvae killing experiment. *gnaA* mutant is affected in CPS synthesis, and thus, influences the *A. baumannii* virulence in *G. mellonella* [47]. Gebhardt et al. have demonstrated that the absence of *ptk*, a gene which codes for CPS export, killed less larvae [48].

*A. baumannii* lipooligosaccharide (LOS) function in bacterial pathogenesis has also attracted attention. LOS is composed of two regions: lipid A moiety and core oligosaccharide. Bartholomew et al. have characterized the lipid A modification by 2-hydroxylation on laurate via LpxO, and tested the survival ability of the corresponding *A. baumannii* mutant. They have shown that LpxO can significantly enhance the survival ability of *A. baumannii* against the innate immune system of larvae [49].

Type VI secretion system (T6SS) is widely distributed in gram-negative bacteria, and can produce and transfer effector molecules into the surrounding environment or neighboring cells [110]. The genes encoding this system in *A. baumannii* have been identified in the genomes of many bacteria, including *A. baumannii*, but T6SS does not affect its potency during *G. mellonella* infection [51]. However, a later study found, for a strain isolated from the environment, that the implication of the T6SS is required for *A. baumannii* to colonize *G. mellonella*, suggesting a strain-dependent process [50].

One of the host’s defense systems used to combat pathogen infections relies on reactive oxygen species (ROS) production. Nevertheless, *A. baumannii* produce superoxide dismutase (SOD) to detoxify the ROS produced by the host. The SOD activity has been analyzed with *G. mellonella*, which has highlighted the importance of SOD during host–pathogen interactions [64].

Nutrients in the host environment are essential for the growth and survival of both host cells and bacterial pathogens. Iron is one of these key micronutrients. In order to prevent oxidative damage caused by free iron in host cells, they are usually isolated in the host by carrier proteins, such as transferrin, lactoferrin, and hemoglobin [111,112]. Bacteria have developed a high-affinity iron acquisition system by the utilization of siderophores, such as acinetobactin, in order to overcome iron sequestration. Zimbler et al. have described that *tonB* mutants, which do not have the ability to provide the energy transduction for iron acquisition, are killing less *G. mellonella* [55]. The research from Gaddy et al. additionally showed that, compared to the wild type, BasD mutant (involved in acinetobactin biosynthesis) and BauA mutant (responsible of acinetobactin transport) produce a lower mortality of *G. mellonella* [52]. These observations indicate a potential use of acinetobactin as a target for therapeutic purposes.

### 4.2. G. mellonella to Study A. baumannii Antibiotic Resistances

The extreme adaptability of *A. baumannii* to antibiotics has allowed this microorganism to develop various resistance mechanisms, and has contributed to the emergence of MDR and even pan-resistant strains worldwide. *G. mellonella* has been established and accepted as one of the in vivo models to explore the *A. baumannii* drug-resistance mechanisms involved in β-lactamases [81], aminoglycoside modifying enzymes [59], and antibiotic target modifications [60].

Β-lactamase is a category of enzymes that can catalyze the hydrolysis and inactivation of β-lactam. According to sequence homology, it can be divided into four classes: class A; B; C; and D [113]. Contrary to class A, C and D enzymes, where the serine residues are catalytically active, the activity of class B enzymes needs to be mediated by zinc and a different heavy metal [114]. Class D β-lactamases (also named oxacillinases (OXAs)) usually hydrolyze carbapenem antibiotics such as isoxazolylpenicillin, oxacillin, and benzylpenicillin, which are commonly used against *A. baumannii* [115]. Tietgen et al. found a novel β-lactamase, OXA-822, isolated from *Acinetobacter calcoaceticus* [81]. The production of OXA-822 was done in *A. baumannii*, and tested upon meropenem treatment in *G. mellonella* infection assays. OXA-822 increases the mortality of infected larvae, indicating carbapenem decreased susceptibility in vivo.

Aminoglycosides are a class of bacterial protein synthesis inhibitors that can interfere with the peptide elongation at the 30S ribosomal subunit [116], therefore affecting bacterial proliferation and growth. Aminoglycoside modifying enzymes (AME), divided into acyltransferase, adenyltransferase, and phosphotransferase, are involved in aminoglycoside resistance [114]. AME genes allow bacterial resistance against amikacin, kanamycin, and tobramycin [116]. Amikacin treatment against resistant strains has been used in combination with peptide/DNA oligomer conjugate, and the efficacy of this new therapy has been tested on *G. mellonella* [59]. The results strongly indicated that this treatment leads to a survival rate comparable to uninfected controls in larvae.

Modifications of antibiotic targets occur in bacteria to escape antibiotics. Lipid A modification from LOS located in the outer membrane is well described in *A. baumannii*. This renders the bacteria resistant to cationic antimicrobial peptides (CAMP) treatment, and protects them from lysis. The addition of galactosamine or phosphoethanolamine (pETN) moiety on lipid A [117] cause colistin resistance, and influence the lipid composition of the bacterial membrane too [118]. LpxMAB, an acyltransferase, is responsible of the addition of two lauric acids (C12:0) on lipid A to form hepta-acylated lipid A. These modifications allow *A. baumannii* to prevent the effect of AMPs released in *G. mellonella* hemolymph [119].

### 4.3. G. mellonella and A. baumannii to Study Bacterial Survival and Spreading

The rapid dissemination of pathogens is a great concern for our society. *G. mellonella* is an interesting tool to monitor the interactions between *A. baumannii* strains and other organisms to give deeper insights into transmission mechanisms, including quorum sensing (QS) [40,61,120], motility [121], and biofilm formation [62].

QS is a well-established mechanism that allows bacteria to sense population density in order to coordinate specific genes expression and group behaviors [122]. In *A. baumannii*, the AbaI inducer and AbaR receptor build the QS circuit [120]. The AbaI/AbaR QS system can enhance *A. baumannii* drug resistance and virulence to *G. mellonella* [120]. Recent studies have identified a third gene, *abaM*, that could regulate the concentration of the QS signal molecule, N-acyl homoserine lactone (AHL) [61]. At the same time, the inactivation of *abaM* leads to an attenuated virulence of *A. baumannii* in the larvae. Oddly, knock-out of the *abaI* gene, which controls the production of the QS signaling molecule, did not alter the lethality of *G. mellonella* larvae [40]. The importance of this gene for *A. baumannii* virulence still needs further investigations.

Five motilities support bacterial movement: swarming; swimming; twitching; gliding; and sliding [123]. Since *A. baumannii* have no flagella [124], fimbriae (type IV pili) have always been considered as the main source of power for bacterial movement through twitching motility. The swarming motility observed with *A. baumannii* should be denominated surface-associated motility, as swarming is flagella dependent [125]. Controlled by *ddc* and *dat* genes, 1,3-diaminopropane (DAP) is an ubiquitous polyamine essential for *A. baumannii* surface-associated motility, and enhances virulence in *G. mellonella* models [63]. A light-regulated type I pilus, mediated by the BlsA photoreceptor, promotes *A. baumannii* surface-associated motility, biofilm formation, as well as virulence reinforcement in the larval model [121].

When planktonic bacteria population colonizes a site, the bacteria secrete extracellular polymeric substances (EPS) to protect them against harsh external environmental changes [126]. Several studies have demonstrated that *A. baumannii* within biofilm are more persistent [127,128]. Using the *G. mellonella* as an in vivo model, Wang et al. revealed that *A. baumannii* from biofilm had higher colistin resistance and stronger virulence than planktonic strains [62].

## 5. Finding New Treatments to Fight *A. baumannii* with *G. mellonella*

With drug resistance increases and virulence evolution, treatment of *A. baumannii* requires important attention. Usually, suitable antimicrobial agents are first screened through in vitro experiments and, later, successful candidates are subjected to in vivo animal assays and, finally, to clinical human validation. In this context, compared with traditional animal models, *G. mellonella* exhibit obvious ethical and logistical advantages. The effectiveness of antimicrobial agents is usually validated within 1–3 days, which saves precious time for the development of new agents [30,129].

### 5.1. Antibiotics

Although the frequent use of antibiotics is the main cause of resistance emergence, they remain nevertheless the dominant treatment strategy due to the lack of viable alternatives. Few effective antibiotic options which combine various therapies appear as promising to cure MDR or pan drug resistant (PDR) *A. baumannii* infections.

Single antibiotic therapy can target pathogens with high selectivity, leading to a better understanding in specific pathogenic mechanisms. With the highly resistant strain, AB5075, rifampin recovery rate was assessed with infected *G. mellonella* larvae [85]. The larval survival rate was 100% at 10 mg/kg of rifampin the first day, and 78% the fourth day. Nishida et al. have evaluated *G. mellonella*-MDR *A. baumannii* infection for the assessment of different antibiotic treatments [76], including colistin, minocycline, polymyxin B, tigecycline, cefozopran, and sitafloxacin. With the same treatment, all the antibiotics had in vivo activity and, remarkably, prolonged the survival rate of the larvae. In consistence with clinical conclusions, colistin can significantly ameliorate the infections caused by *A. baumannii* (cured cases/total infected cases:156/198, 79%) [130]. In addition to evaluating the efficacy, *G. mellonella* models can also be successfully used to detect the toxicity of antibiotics. For example, Cruz-Muñiz et al. have estimated the toxicity and antibiotic activity of mitomycin C in non-infected and infected larval models [82]. The results showed a 100% survival rate of non-infected larvae, and more than 50% of larvae infected with three different MDR *A. baumannii* strains, indicating the safety and efficiency of mitomycin C to cure *A. baumannii* invasion.

Antibiotic combination therapies can act synergistically, and, therefore, lead to pathogen clearance acceleration [131]. Colistin, as the last resort to combat the MDR bacteria, has been the most popular option in *A. baumannii* treatment. However, overuse of colistin has caused the gradual increase of the minimal inhibitory concentration (MIC), and colistin-resistant *A. baumannii* have been identified and characterized [132]. Therefore, colistin combination treatments are in use, such as vancomycin/colistin [72,87], teicoplanin/colistin [72], daptomycin/colistin [86], levofloxacin/colistin [74], cotrimoxazole/colistin [75], etc. Hornsey et al. illustrated the efficacy of telavancin/colistin for dramatically improving the survival of *A. baumannii* infected larvae compared to telavancin alone or colistin alone [73]. Likewise, O’Hara et al. demonstrated the benefits of multiple antibiotic combination therapy (doripenem/vancomycin/colistin) [79]. Corresponding to clinical cases, colistin combination therapy can reduce the risk of nephrotoxicity compared with monotherapy [133]. Although combined treatments are more effective, their mechanisms of action need better understanding to avoid the development of even more resistant strains.

### 5.2. Further Strategies

The development of new antimicrobial agents is necessary to fight MDR *A. baumannii* diseases. Testing these unconventional antimicrobial agents with an in vivo model, such as *G. mellonella*, can be very helpful to analyze their activity and toxicity.

Bacteriophage therapy is an alternative approach owing to its potential advantages to target MDR bacteria with high specificity and selectivity. Furthermore, it is easily available and a safe therapeutic modality for immunocompetent and immunocompromised patients [134,135]. Phage treatment of *A. baumannii* infections has been broadly explored. Jeon et al. have analyzed the effectiveness of targeting carbapenem-resistant *A. baumannii* with lytic phages (Βϕ-R2096) in *G. mellonella* models [11]. After 48 h post-infection, they obtained 100% and 50% (MOI = 100) survival rate for non-infected and infected larvae, respectively. Histological results showed that the non-infected group did not exhibit any tissue damage, whereas the infected group had an obvious reduction of tissue damage and fat body melanization. Similar results were also obtained with extensively drug-resistant *A. baumannii* strains [80]. Additionally, phage-based combined therapy, such as phage/polymyxin B, phage/meropenem, phage/ciprofloxacin, and phage/gentamicin, demonstrated excellent results against *A. baumannii* infection in larval models [77,88]. Moreover, these therapy strategies are gradually approaching broad clinical practice [136,137].

Furthermore, natural antibacterial agents might be possible tools to treat human health problems caused by *A. baumannii* infections. Studies showed no melanization after injection at any concentration with polyphenols, theaflavin, and epicatechin [90]. This result suggests the efficacy of these polyphenols against *A. baumannii* infection in vivo.

*G. mellonella* has also been employed to investigate the antibiotic activity of novel therapeutic strategies based either on metal or non-metal compounds, such as manganese(I) tricarbonyl complexes [93], silver acetate [95], gallium nitrate [91], gallium protoporphyrin IX [92], and homodimeric tobramycin adjuvant [84].

## 6. Conclusions

Over recent years, *G. mellonella* appeared as a powerful, reliable, fast, and cheap host–pathogen infection model, and a good alternative host to study *A. baumannii* virulence and new antimicrobial agent efficacy. Although it cannot replace mammalian models, the initial data collected through *G. mellonella* assays provide an important reference for new drug development and clinical applications. However, many teams have noticed that different prominent parameters may impact *A. baumannii* infection effects. Therefore, standardized regulations, such as control of the inoculum dose, temperature, or incubation time, are very important for the study of the *G. mellonella*-*A. baumannii* model. Moreover, with *G. mellonella* genome sequence availability, associated with new molecular tools, this insect model will be precious for future biomedical researches [138].

## Figures and Tables

**Figure 1 pathogens-10-01483-f001:**
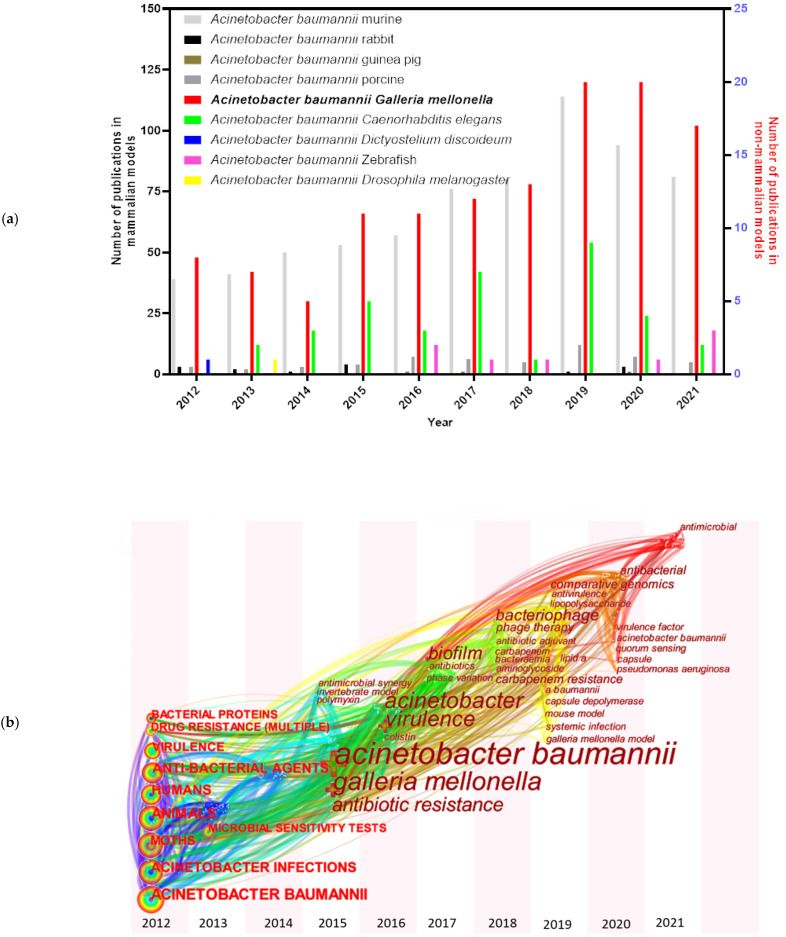
Pubmed literature review focused on *A. baumannii* and animal models. (**a**) Number of publications about *A. baumannii* associated with mammalian and non-mammalian models: “*Acinetobacter baumannii* murine” (685); “*Acinetobacter baumannii* rabbit” (14); “*Acinetobacter baumannii* guinea pig” (3); “*Acinetobacter baumannii* porcine” (54); “*Acinetobacter baumannii Galleria mellonella*” (124); “*Acinetobacter baumannii Caenorhabditis elegans*” (36); “*Acinetobacter baumannii Dictyostelium discoideum*” (1); “*Acinetobacter baumannii* zebrafish” (8); and “*Acinetobacter baumannii Drosophila melanogaster*” (1) on Pubmed over the period Jan 2012 to Sep 2021. Note: “query term on Pubmed” (total number of publications). (**b**) Distribution map of keywords and nodes time-zone associated with *G. mellonella* and *A. baumannii*.

**Figure 2 pathogens-10-01483-f002:**
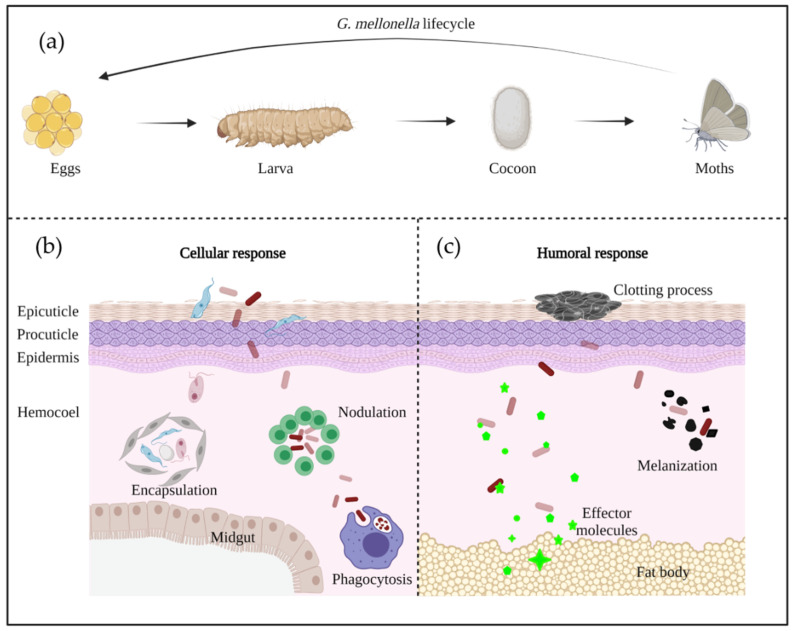
The life cycle (**a**) and immune system (**b**,**c**) of *G. mellonella*.

**Figure 3 pathogens-10-01483-f003:**
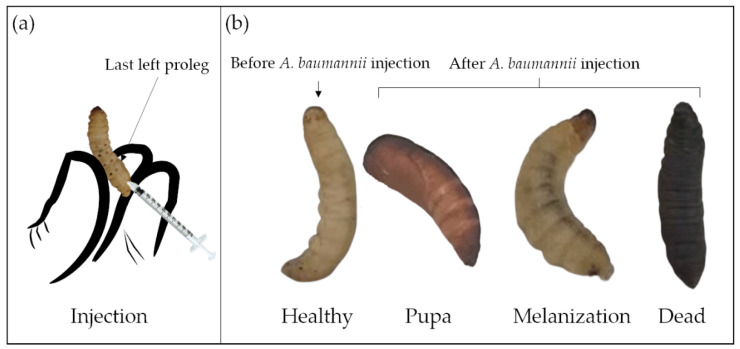
Injection model (**a**) and different health states (**b**) of *G. mellonella*.

**Table 1 pathogens-10-01483-t001:** Protocols analyzing the pathogenicity of *A. baumannii* in *G. mellonella*.

*A. baumannii*			Larva/ Group	Larva Inoculation			Larva Incubation		Refs
Pathogenicity		Strains and Mutants		Style	Volume/Larva	Concentration	Temp	Time	
Virulence factors									
Phospholipases	Phospholipases C	Δ*plcN*	20	Injection	10 µL	2 × 10^6^ CFU/mL	37 °C	8 days	[42,43]
		ATCC 19606^T^, *plc2::aph, plc1::aph-FRT, plc1::ermAM/plc2::aph*	10	Injection		1 × 10^5^ CFU	37 °C	5 days	
	Phospholipases D	ATCC 19606^T^, Δ*pld*	16	Injection	10 µL	1 × 10^6^ CFU/mL	37 °C	4 days	[44]
Membrane proteins	Surface antigen protein 1 (SurA1)	ATCC 17978, CCGGD201101, Δ*SurA1*	20	Injection	20 µL	1 × 10^6^ CFU/mL	37 °C	7 days	[45]
Capsular polysaccharides and LOS	Capsule genes, *epsA* and *ptk*	AB5075, AB5075 *epsA*::Tn5, AB5075 *ptk*::Tn5	-	Injection	5 µL	1 × 10^7^ CFU/mL	37 °C	5 days	[46]
	K locus	MDR-ZJ06, Δ*gnaA*	-	Injection	10 µL	1 × 10^8^ CFU/mL	37 °C	3 days	[47]
	*ptk* gene	AB5075, Δ*ptk*	-	Injection	-	1 × 10^5^, 1 × 10^6^ CFU	37 °C	6 days	[48]
	LOS	ATCC 17978, Δ*lpxO*, Δ*lpxO::*Tn7*lpxO*	10	Injection	10 µL	5 × 10^4^ CFU	37 °C	3 days	[49]
Protein secretion system	Type VI secretion system (T6SS)	DSM30011, Δ*tssM*	20	Injection	10 µL	1 × 10^5^ CFU	37 °C	-	[50,51]
		17978, 17978 Δ*tssM*	10	Injection	5 µL	10^6^–10^7^ CFU	37 °C	40–60 h	
Metal acquisition systems	Iron acquisition	ATCC 19606^T^, *basD*, *bauA*	30, 10	Injection	5 µL	1 × 10^2^, 1 × 10^5^ CFU	37 °C	18 h/6 days	[52,53,54,55]
		A118, ATCC 19606^T^, ATCC 17978	-	Injection	-	1 × 10^5^ CFU	37 °C	6 days	
		ATCC 19606^T^, Δ*basD*	30	Injection	10 µL	OD_600_: 0.2	37 °C	72 h	
		ATCC 19606^T^, *entA::aph, tonB1::aph, tonB2::aacC1, tonB1::aph tonB2::aacC1*	10	Injection	-	1 × 10^5^ CFU	37 °C	6 days	
	Zinc acquisition	AB5075, *znuB::Tn*	-	Injection	-	1 × 10^6^ CFU	37 °C	0 h, 4 h	[48]
Antimicrobial resistance									
β-lactamases		AB5075, ZJ06, LS01, ATCC 17978	10	Injection	10 µL	OD_600_: 0.1	37 °C	72 h	[56]
Efflux pumps		ATCC 17978, *A1S*	16	Injection	10 µL	OD_600_: 0.5	37 °C	96 h	[57]
Permeability defects		ATCC 19606, Δ*kup*Δ*trk*Δ*kdp*, Δ*kup*Δ*trk*	20	Injection	10 µL	1 × 10^6^ CFU	37 °C	6 days	[58]
Aminoglycoside modifying enzymes		*Ab*A155	10	Injection	5 µL	5 × 10^5^ CFU	37 °C	>120 h	[59]
Alternation of target sites		MB_2, MB_6C, MB_23C, MB_177, MB_90, MB_119, SG3161, SG3166	10	Injection	-	1 × 10^5^ CFU	37 °C	96 h	[60]
Dissemination									
Quorum sensing	3-hydroxy-C12-homoserine lactone	M2, *aba1*::Km	16	Injection	10 µL	>0.5 log CFU	37 °C	6 days	[40]
	*abaM* gene	AB5075, *abaI*::T26, *abaM*::T26	10	Injection	-	2 × 10^4^ CFU, 2 × 10^5^ CFU	37 °C	120 h	[61]
Biofilm		NCTC 12156, NCTC 10303, ATCC 17978, NCTC 13302, W1, NCTC 13423, ATCC BAA-1710, NCTC 13424, ATCC BAA-1709, UKA1-UKA19	10	Injection	-	1 × 10^5^, 1 × 10^6^ CFU	37 °C	5 days	[62]
Motility		ATCC 17978, 129/*ddc*, 277/*dat*	16	Injection	5 µL	3 × 10^5^ CFU	37 °C	5 days	[63]
Others									
Stress response	Reactive oxygen species (ROS) resistance	ATCC 17978, ATCC 17978 *sod2343::Km, ATCC 17978 sod2343::Km pWHsod2343*	16, 10	Injection	5 µL	3 × 10^5^ CFU, 1.5 × 10^6^ CFU	37 °C, −80 °C	5 days, immediately	[64]
	Temperature	ATCC 17978	-	Injection	10 µL	1 × 10^6^ CFU/mL	28 °C, 37 °C	72 h	[65]
	Ethanol	ATCC 19606^T^	30	Injection	-	1 × 10^5^ CFU	37 °C	6 days	[66]
Phase-variable switch		AB5075 opaque, AB5075 translucent	10	Injection	-	3 × 10^4^ CFU	37 °C	24 h	[67,68]
		AB5075, Δ*ompR*, Δ*envZ*, Δ*ompR* Δ*envZ*	30	Injection	-	10^3^–10^4^ CFU	37 °C	5 days	

**Table 2 pathogens-10-01483-t002:** The *G. mellonella* infection model for screening prospective treatment options against *A. baumannii*.

Category	*A. baumannii*	Treatment Type	Dose		Time	Refs
			Volume/Larva	Concentration		
AMPs						
Amphiphilic peptide zp3	-	Post-treatment	10 µL	200–800 mg/kg	30 min	[69]
Anti-*lpxB* pPNA	MDR	Post-treatment	10 µL	75 mg/kg	1 h	[70]
PNA (RXR)4 XB	MDR	Post-treatment	10 µL	150/600 µM	30 min	[71]
Antibiotics						
Colistin	MDR	Post-treatment	10 µL	2.5 mg/kg	30 min	[70,72,73,74,75,76]
	-	Post-treatment	10 µL	2.5 mg/kg	30 min	
	Clinical isolate	Post-treatment	10 µL	2.5 mg/kg	2 h	
	Carbapenem-resistant	Post-treatment	10 µL	2.5 mg/kg	2 h	
	Colistin-resistant	Post-treatment	5 µL	2.5 mg/kg	30 ± 5 min	
	MDR	Post-treatment	10 µL	40 mg/kg	-	
	MDR	Post-treatment	10 µL	2 mg/kg	1 h	
Cefozopran	MDR	Post-treatment	10 µL	40 mg/kg	-	[76]
Ciprofloxacin	-	Post-treatment	-	10 mg/kg	20 min	[77]
Clarithromycin	MDR	Pre-treatment	5 µL	25 mg/kg	2.5 h	[78]
Cotrimoxazole	Carbapenem-resistant	Post-treatment	10 µL	10 mg/kg	2 h	[75]
Doripenem	Colistin-resistant	Post-treatment	5 µL	7.5 mg/kg	30 ± 5 min	[79]
Gentamicin	-	Post-treatment	-	8 mg/kg	20 min	[72,77]
	-	Post-treatment	-	8 mg/kg	20 min	
Imipenem	MDR	Post-treatment	-	5 mg/mL	30 min	[80]
Levofloxacin	MDR	Post-treatment	10 µL	6.7 mg/kg	2 h	[74]
Meropenem	Clinical isolate	Post-treatment	10 µL	4 mg/kg	1 h	[77,81]
	-	Post-treatment	-	20 mg/kg	20 min	
Minocycline	MDR	Post-treatment	10 µL	40 mg/kg	-	[76]
Mitomycin C	-	Post-treatment	-	13–16 mg/kg	2–5 min	[82]
Netropsin	Clinical isolate	Post-treatment	5 µL	12.5 mg/L	30 min	[83]
Novobiocin	MDR	Post-treatment	10 µL	100 mg/kg	3 h	[84]
Polymyxin B	Clinical isolate	Post-treatment	5 µL	4 mg/L	30 min	[76,83]
	MDR	Post-treatment	10 µL	40 mg/kg	-	
Rifampicin	MDR	Post-treatment	2 µL	2.5, 5, 10 mg/kg	30 min	[85]
Sitafloxacin	MDR	Post-treatment	10 µL	40 mg/kg	-	[76]
Teicoplanin	MDR	Post-treatment	10 µL	10 mg/kg	30 min	[72]
Telavancin	-	Post-treatment	10 µL	10 mg/kg	30 min	[73]
Tetracycline	MDR	Post-treatment	10 µL	40 mg/kg	-	[76]
Tigecycline	MDR	Post-treatment	10 µL	40 mg/kg	-	[76]
Vancomycin	Colistin-resistant	Post-treatment	5 µL	15 mg/kg	30 ± 5 min	[79]
Cotrimoxazole/colistin	Carbapenem-resistant	Post-treatment	10 µL	10 mg/kg + 2.5 mg/kg	2 h	[75]
Daptomycin/colistin	MDR	Post-treatment	-	4 mg/L + 2.5 mg/L	2 h	[86]
Doripenem/Vancomycin	Colistin-resistant	Post-treatment	5 µL	7.5 mg/kg + 15 mg/kg	30 ± 5 min	[79]
Doripenem/Vancomycin/colistin	Colistin-resistant	Post-treatment	5 µL	7.5 mg/kg + 15 mg/kg + 2.5 mg/kg	30 ± 5 min	[79]
Levofloxacin/colistin	MDR	Post-treatment	10 µL	6.7 mg/kg + 2.5 mg/kg	2 h	[74]
Polymyxin B/netropsin	Clinical isolate	Post-treatment	5 µL	4 mg/L + 12.5 mg/L	30 min	[83]
Teicoplanin/colistin	MDR	Post-treatment	10 µL	10 mg/kg + 2.5 mg/kg	30 min	[72]
Telavancin/colistin	-	Post-treatment	10 µL	10 mg/kg + 2.5 mg/kg	30 min	[73]
Vancomycin/colistin	MDR	Post-treatment	10 µL	15 mg/kg + 2.5 mg/kg	2 h	[72,87]
	MDR	Post-treatment	10 µL	10 mg/kg + 2.5 mg/kg	30 min	
Others						
Anti-*lpxB* pPNA/colistin	MDR	Post-treatment	10 µL	75 mg/kg + 2 mg/kg	1 h	[70]
Bacteriophage	Carbapenem-resistant	Post-treatment	5 µL	1 × 10^10^, 1 × 10^9^ PFU/mL	30 min	[11,77,80,88]
	-	Post-treatment	-	MOI ≈ 1	20 min	
	MDR	Post-treatment	10 µL	5.10^7^ PFU, MOI = 100	30 min	
	Carbapenem-resistant	Post-treatment	10 µL	10^4^ pfu	30 min	
Capsule depolymerase Dpo48	Extensive drug-resistant	Pre-treatment, post-treatment	10 µL	50 µg/mL, 5 µg	1 h, 5 min	[89]
Epicatechin	MDR	Post-treatment	-	40 mg/kg	30 min	[90]
Homodimeric Tobramycin Adjuvant/Novobiocin	MDR	Post-treatment	10 µL	25/50 mg/kg + 25/50 mg/kg	3 h	[84]
Gallium nitrate	MDR	Post-treatment	-	1.2 mmol/kg	15 min	[91]
Gallium protoporphyrin IX	-	Simultaneously	5 µL	20, 40 µg/mL	-	[92]
Manganese (i) tricarbonyl complexes	MDR	Post-treatment	-	5 mg/kg	30 min	[93]
SCH-79797	MDR	Simultaneously		66.6 µg/larva	-	[94]
Silver acetate	Carbapenem-resistant	Post-treatment	-	0, 10, 20 mg/kg	30 min	[95]
Theaflavin	MDR	Post-treatment	-	20 mg/kg	30 min	[90]
Theaflavin/Epicatechin	MDR	Post-treatment	-	20 mg/kg + 40 mg/kg	30 min	[90]
Bacteriophage/Ciprofloxacin	-	Post-treatment	-	MOI ≈ 1 + 10 mg/kg	20 min	[77]
Bacteriophage/Gentamicin	-	Post-treatment	-	MOI ≈ 1 + 8 mg/kg	20 min	[77]
Bacteriophage/Meropenem	-	Post-treatment	-	MOI ≈ 1 + 20 mg/kg	20 min	[77]
Endolysin/colistin	-	Post-treatment	10 µL	25 µg/mL + 1/4 MIC	1 h	[96]

**Notes:** MDR: multi-drug resistant. Pre-treatment/post-treatment: the antimicrobial agents were added before/after the *A. baumannii* infection. MOI: multiplicity of infection. CFU: colony forming unit. Time: the period between the first and second injection.

**Table 3 pathogens-10-01483-t003:** Health index scoring system of *G. mellonella* adapted from [101].

Category	Description	Score
Activity	No movement	0
	Minimal movement on stimulation	1
	Move when stimulated	2
	Move without stimulation	3
Cocoon formation	No cocoon	0
	Partial cocoon	0.5
	Full cocoon	1
Melanization	Black larvae	0
	Black spots on brown larvae	1
	≥3 spots on beige larvae	2
	<3 spots on beige larvae	3
	No melanization	4
Survival	Dead	0
	Alive	2

## Data Availability

Not applicable.

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
