# Peer review of "Galleria mellonella as a Good Model to Study Acinetobacter baumannii Pathogenesis"

_pathogens, 2021, doi:10.3390/pathogens10111483_

Round 1

Reviewer 1 Report

This review is comprehensive and it is well written. There are comments for more improvement:

1- Line 19: In vivo, needs to be in italic, and in other places throughout the review

2- it will be helpful if the authors provide the methods of injection of Wax moth with providing pictures for these methods 

3- Table 1: provide more columns and include the references for each study

4- Be sure that G. mellonella and A. baumannii, in vitro  are in italic throughout the manuscript including the titles

5- It is recommended to add a new section in this review comparing the G. mellonella to other in vivo models 

Reviewer 2 Report

Dear Authors,

Thank you for the opportunity to read this interesting work on the use of Galleria mellonella as a model to study Acinetobacter baumannii pathogenesis. The discussed issue has been presented extensively with the use of the latest available literature. Figure 1 deserves special attention, as it provides an interesting Pubmed literature review.

I present some of my comments on the manuscript below:

Lines 34-36 There is also a report on the use of the porcine model in Abaumannii research. Maybe it's worth considering as well. Here is this publication: DOI: 10.1089/wound.2018.0786

Lines 100-102 In my opinion, it is worth mentioning the bacteria and fungi used in the research with G. mellonella as a model. For example, the most commonly used fungi are:

Conidiobolus coronatus: 10.1371/journal.pone.0228556, 10.1371/journal.pone.0228407, 10.1186/s13578-019-0291-1, 10.1038/s41598-021-95440-6

Beauveria bassiana: 10.1007/s00203-017-1456-0, 10.1111/1744-7917.12706, 10.1111/1744-7917.12706

Metarhizium species: 10.1111/1744-7917.12706, 10.1080/21505594.2017.1405190, 10.1080/21505594.2019.1693230

Lines 119-122 There is no literature reference

Table 2. I did not find an extension of the MDR abbreviation. My guess is that this means 'multidrug-resistant', but the lack of explanation may mislead the reader.

Fig 3. In my opinion, the photos are very blurry and have low resolution. Is it possible to improve these photos? Are these photos of insects after A. baumannii infection?

Round 2

Reviewer 1 Report

Thanks you so much for addressing all my previous comments